# Systems Genomics Reveals microRNA Regulation of ICS Response in Childhood Asthma

**DOI:** 10.3390/cells12111505

**Published:** 2023-05-29

**Authors:** Rinku Sharma, Anshul Tiwari, Alvin T. Kho, Juan C. Celedón, Scott T. Weiss, Kelan G. Tantisira, Michael J. McGeachie

**Affiliations:** 1Channing Division of Network Medicine, Brigham and Women’s Hospital, Harvard Medical School, Boston, MA 02115, USA; 2Department of Molecular Physiology and Biophysics, Vanderbilt University, Nashville, TN 37235, USA; 3Computational Health Informatics Program, Boston Children’s Hospital, Boston, MA 02115, USA; 4Division of Pediatric Pulmonary Medicine, UPMC Children’s Hospital of Pittsburgh, University of Pittsburgh, Pittsburgh, PA 15260, USA; 5Division of Pediatric Respiratory Medicine, University of California San Diego, Rady Children’s Hospital, San Diego, CA 92123, USA

**Keywords:** childhood asthma, miRNA, ICS

## Abstract

Background: Asthmatic patients’ responses to inhaled corticosteroids (ICS) are variable and difficult to quantify. We have previously defined a Cross-sectional Asthma STEroid Response (CASTER) measure of ICS response. MicroRNAs (miRNAs) have shown strong effects on asthma and inflammatory processes. Objective: The purpose of this study was to identify key associations between circulating miRNAs and ICS response in childhood asthma. Methods: Small RNA sequencing in peripheral blood serum from 580 children with asthma on ICS treatment from The Genetics of Asthma in Costa Rica Study (GACRS) was used to identify miRNAs associated with ICS response using generalized linear models. Replication was conducted in children on ICS from the Childhood Asthma Management Program (CAMP) cohort. The association between replicated miRNAs and the transcriptome of lymphoblastoid cell lines in response to a glucocorticoid was assessed. Results: The association study on the GACRS cohort identified 36 miRNAs associated with ICS response at 10% false discovery rate (FDR), three of which (miR-28-5p, miR-339-3p, and miR-432-5p) were in the same direction of effect and significant in the CAMP replication cohort. In addition, in vitro steroid response lymphoblastoid gene expression analysis revealed 22 dexamethasone responsive genes were significantly associated with three replicated miRNAs. Furthermore, Weighted Gene Co-expression Network Analysis (WGCNA) revealed a significant association between miR-339-3p and two modules (black and magenta) of genes associated with immune response and inflammation pathways. Conclusion: This study highlighted significant association between circulating miRNAs miR-28-5p, miR-339-3p, and miR-432-5p and ICS response. miR-339-3p may be involved in immune dysregulation, which leads to a poor response to ICS treatment.

## 1. Introduction

Asthma is the most common chronic lung disease of childhood, with 44.3 percent of children in the United States reporting one or more asthma attacks in the previous year [1]. Inhaled corticosteroids (ICS) are the most effective and widely recommended controller medication for asthma. However, there are two key drawbacks: over 40% of patients do not respond well to ICS therapy, particularly those with severe disease, and ICS have dose-dependent side effects [2,3,4]. Clinical asthma management would benefit from a method to identify patients who are poor responders to ICS before treatment begins, allowing other medications to be added or substituted for the ICS and avoiding the usual trial-and-error phase of treatment. Extracellular microRNAs (miRNAs) are miRNAs found outside of cells, in bodily fluids such as saliva, urine, breast milk, etc. They are released from cells through various mechanisms, such as passive leakage from damaged cells, active secretion in extracellular vesicles (EVs), or by binding to proteins such as Argonaute-2 (AGO2). The release of extracellular miRNAs is more localized and may affect cells that are in close proximity to the site of release. In contrast, circulating miRNAs are a type of extracellular miRNA that circulate in the bloodstream and have the potential to regulate gene expression in distant cells. They are mainly released by active secretion in extracellular vesicles or by binding to proteins such as high-density lipo-proteins (HDLs), apoptotic bodies, and micro-particles [5]. Circulating microRNAs (miRNAs) have been suggested as biomarkers for a number of conditions and have been shown to be important in a number of inflammatory-mediated processes [6,7], including asthma [8]. We hypothesized that circulating miRNA in serum would be associated with ICS response in children with asthma.

To distinguish between good and poor ICS responders, a precise quantitative definition of steroid response is necessary. Six clinical features were used by Clemmer et al. to establish a measure of the Steroid Responsiveness Endophenotype (SRE); while this method showed excellent performance, this measure is inherently longitudinal and necessitates surveillance during a period of ICS administration [9]. We have previously defined a Cross-sectional Asthma STEroid Response (CASTER) measure of ICS response, based on a combination of asthma control indicators and spirometry measures, which can be computed from data collected at a single time point [10]. Previously, analyses of gene or miRNA differential expression [11,12], metabolome, single-nucleotide polymorphism, and expression quantitative trait loci (eQTL) found multiple genes, miRNAs, metabolomes, or SNPs associated with asthma drug responsiveness [13,14,15,16].

The primary goal of this study was to identify circulatory miRNAs associated with ICS response in children with asthma and to interpret the mode of function of such miRNAs. We show that serum miRNAs were associated with ICS response in a cohort of Costa Rican children with asthma, and these findings were replicated in an independent cohort of North American children with asthma. Using transcriptomics of lymphoblastoid cell lines in response to steroids, we then constructed a steroid-responsive gene co-expression network, identified co-modulated genes clusters using a modified WGCNA method, and showed that these were associated with replicated miRNAs.

## 2. Materials and Methods

### 2.1. Study Cohorts: GACRS and CAMP

#### 2.1.1. Discovery Cohort: The Genetics of Asthma in Costa Rica Study (GACRS)

The Genetics of Asthma in Costa Rica Study (GACRS) is a cross-sectional study that recruited from February 2001 to August 2008. It included 1165 asthmatic children between the ages of 6 and 14 years, each with a high probability of having six or more great-grandparents from the Central Valley of Costa Rica. A previous publication [17] described the GACRS protocols and assessments, which included FEV1, bronchodilator response (BDR), methacholine challenge, and questionnaires about prior medication use and exacerbations. GACRS did not specifically assess asthma severity. We used 2 major types of data from selected GACRS participants: clinical data (from which we calculated CASTER), and serum miRNA data.

#### 2.1.2. Replication Cohort: The Childhood Asthma Management Program (CAMP)

CAMP is a multi-center, randomized, double-blinded, clinical trial of inhaled corticosteroids in 1041 children aged 5 to 12 years with mild-to-moderate persistent asthma who were followed for four years. In CAMP, asthma severity was defined at enrollment based on physician assessment (1 = no asthma, 2 = mild, and 3 = moderate asthma), and patients with mild and moderate asthma were enrolled in the cohort. A detailed definition is given in Saprio et al. 1999 [18]. CAMP participants predominantly self-identified as non-Hispanic white, but included small numbers of African American, Hispanic white, and other racial and ethnic groups (Table 1), which were grouped together in this analysis. The design [18] and outcomes [19] of the CAMP study have been previously reported (“other”). During the original trial, CAMP participants were randomized into 3 treatment arms: budesonide (an ICS), nedocromil, or placebo. We used 3 major types of data from selected CAMP participants: clinical data (from which we calculated CASTER), serum miRNA data, and transcriptomic profiling from lymphoblastoid cell lines.

### 2.2. CASTER: Cross-Sectional Asthma STEroid Response Measurement

Cross-sectional Asthma STEroid Response (CASTER) is a composite corticosteroid responsiveness phenotype or Steroid Responsiveness Endophenotype (SRE) measure. CASTER is calculated using five cross-sectional clinical measures: (1) oral steroid courses, (2) asthma-related hospitalizations and ED (emergency department) visits, (3) pre-bronchodilator FEV1, (4) provocative concentration of methacholine required to effect a 20% reduction of FEV1 (PC20), and (5) bronchodilator response to albuterol (BDR), computed as a percentage change in FEV1 from baseline. In previous work, we showed that CASTER had good performance in childhood cohorts [10].

### 2.3. miRNA Sequencing and Quality Control

We performed small RNA sequencing on serum from 580 GACRS samples of asthma patients on ICS. Sequencing on 187 CAMP samples has been described previously [20]. Both cohorts were sequenced following the same protocols [20]. In brief, small RNA-seq libraries were prepared by using the Norgen Biotek Small RNA Library Prep Kit v2 (Norgen Biotek, Therold, ON, Canada) and sequenced on the Illumina NextSeq 500 platform. The ExceRpt pipeline was used for quality control (QC) of the RNA-seq data [21]. miRNAs with less than five mapped reads in at least 50% of subjects were removed. The data were normalized using DESeq2 [22].

Small RNA sequencing was completed on all available serum samples from GACRS [*n* = 1134]. Each sample produced an average of 15.8 million total reads. In total, 11.9 million reads per sample passed the initial quality control (QC). On average, 8.9 million reads per sample were mapped to the genome, which included 5.4 million miRNA sense sequences.

DESeq2 was used to perform normalization before association analysis. For GACRS cohort samples, sequencing was carried out in 2 batches, which may have introduced technical causes of discrepancy during preparation and handling, affecting the outcomes. Therefore, guided principal component analysis (gPCA) [23] was used to check for batch effects on mapped read counts per sample; the analysis showed that there was not a significant batch effect in normalized data (*p*-value = 0.41) (Appendix A).

### 2.4. ICS-Response-Associated miRNA Identification

The CASTER measure of ICS response was computed for all subjects in GACRS and separately for CAMP. For more details, see [10], where partial least squares (PLS) distance and principal component (PCA) methods were used to classify good vs. poor ICS responders. We used mean (PCA-based) CASTER values from each cohort to dichotomize participants into poor and good responder groups. The patients having a CASTER value greater than the mean value of their cohort were considered to have a good response to ICS treatment, while those with a CASTER value less than the mean value were considered to have a poor response. Logistic regression with a Benjamini–Hochberg false discovery rate (FDR) correction for multiple testing was used to identify miRNAs associated with ICS response in GACRS using the GLM function in stats (v 4.1.2) R package. A significance threshold of 10% FDR was used. The analysis was adjusted for age and sex. Similarly, logistic regression was used to assess the association of serum miRNAs with ICS response using CASTER phenotype in CAMP while adjusted for age, sex, and race or ethnicity (non-Hispanic white, Hispanic, and others). Sensitivity analysis in CAMP was also performed, including asthma severity and genetic ancestry proportion as co-variates. A *p*-value < 0.05 was considered significant for replication in the CAMP cohort.

Logistic regression was also used to define a predictive model of ICS response. Three models were assessed in this study: Model1 included demographic variables (age, sex, race/ethnicity, height, weight, BMI), clinical variables (log10 IgE, vitamin, log10 Eosinophil count), smoking, and asthma severity; Model2 consisted of three miRNAs that were replicated; and Model3 combined variables from Model1 and Model2. These models were trained on the discovery cohort, GACRS, and were then evaluated on the replication cohort (CAMP) without parameter refitting. The area under the receiver operator characteristic curve (AUC) was used to measure predictive accuracy, and the AUC confidence interval was measured using 1000 bootstrap. The three models were compared using ANOVA.

### 2.5. CAMP Lymphoblastoid Cell Lines (LCL) Transcriptome Data

Some CAMP subjects also had gene array expression data available from a later timepoint. As previously described [14,15], subjects provided blood samples from which CD41 lymphocytes were isolated. These samples were immortalized and transcriptomically profiled under two treatment conditions: dexamethasone (DEX) (1026 mol/L) and a sham [ethanol] control. After 6 h, mRNA expression levels were measured with the Illumina HumanRef8 v2 BeadChip (Illumina, San Diego, CA, USA). Detailed protocols were previously described [14]. We performed quality control (QC) and filtering processes for the present study as described previously [15]. For this study, we looked at 88 samples of individuals on ICS for which serum miRNA data was also available.

### 2.6. Functional Annotation of Replicated miRNAs

The gene targets for three replicated CASTER-associated miRNAs were identified by multiMiR R package v 1.12 [24], where miRecords v 4, TarBase v 8, and miRTarBase v 7.0 databases were used to provide validated mRNA targets. These databases contain lists of validated miRNA target genes identified using experimental methods such as luciferase reporter assay, HITS-CLIP, CLASH, qRT-PCR, Western blot, degradome sequencing, immunocytochemistry, and others. The Database for Annotation, Visualization and Integrated Discovery (DAVID) v 2021 [25] was used to perform functional enrichment analysis of total unique targets of 3 replicated miRNAs, where gene ontology (molecular function and biological process) and pathway (KEGG and Reactome) datasets were considered. To show substantial enrichment of targeted genes for a pathway, we used a Bonferroni-adjusted *p*-value cut-off of 0.10 and a gene count of 3 or more.

### 2.7. Gene Expression Analysis

To identify differentially expressed genes in response to dexamethasone (DEX), gene expression from LCL cell lines was examined using the limma R package [26]. The association between three replicated miRNAs and differentially expressed genes in response to dexamethasone was then investigated using linear mixed modeling, using the lme4 R package [27]. We tested whether the direct targets of three miRNAs were over-represented among genes differentially expressed in response to DEX treatment using the hypergeometric test available in the stats R package. We identified comodulated genes (modules) using Weighted Gene Correlation Network Analysis (WGCNA) v 1.71 [28]. For the WCGNA input matrix, we used a modified version of Pearson correlation, following [29], to account for the correlation between the DEX and sham paired expression design. We then assessed the association of each module’s eigengene (ME) to miRNA levels, where the module eigengene is defined as the first principal component of the module and represents the overall expression level of the module. To account for the within-pair correlation in data from the DEX–sham samples’ paired design, we used a linear mixed-effects model (LMM) to test the association of a gene module to miRNA, with adjustment for age and sex using the lme4 R package [27]. We considered 10% FDR cutoff as a significance threshold.

## 3. Results

### 3.1. Cohort Characteristics

#### 3.1.1. GACRS

Small RNA-Seq data from the baseline serum were available for 1134 (97%) of the 1165 children in the GACRS. A total of 580 GACRS participants (51%) who were self-reported to use ICS in the prior 6 months and had enough data to calculate their CASTER phenotype were divided into two groups: those with CASTER values below the mean CASTER value (0.00403) were classified as poor responders (*n* = 379), and those with CASTER values above the mean were classified as good responders (*n* = 201) (Table 1).

**Table 1 cells-12-01505-t001:** Baseline Epidemiologic and Clinical Characteristics of the GACRS and CAMP cohort data.

	GACRS			CAMP		
Characteristics	Poor Responder	Good Responder	*p* Value	Poor Responder	Good Responder	*p* Value
	(*n* = 379)	(*n* = 201)		(*n* = 71)	(*n* = 116)	
Sex						
Male	156 (41.2%)	87 (43.3%)	0.686	39 (54.9%)	70 (60.3%)	0.565
Female	223 (58.8%)	114 (56.7%)		32 (45.1%)	46 (39.7%)	
Age (years)						
Mean (SD)	9.43 (1.89)	8.80 (1.87)	<0.001	8.67 (2.18)	9.16 (2.08)	0.132
Median [Min, Max]	9.36 [4.50, 15.2]	8.53 [6.02, 13.1]		8.90 [5.18, 13.2]	8.94 [5.26, 13.1]	
Race/Ethnicity						
Non-Hispanic	NA	NA		50 (70.4%)	76 (65.5%)	0.211
Hispanic White	379 (100%)	201 (100%)		4 (5.6%)	16 (13.8%)	
Other	NA	NA		17 (23.9%)	24 (20.7%)	
Height (cm)						
Mean (SD)	134 (15.2)	130 (12.3)	<0.001	132 (12.9)	136 (13.7)	0.0615
Median [Min, Max]	134 [0, 168]	128 [103, 164]		133 [108, 156]	136 [107, 170]	
Missing	NA	NA		1 (1.4%)	0 (0%)	
Weight (kg)						
Mean (SD)	34.5 (11.9)	32.1 (12.3)	0.0275	32.8 (12.0)	35.6 (12.5)	0.133
Median [Min, Max]	30.9 [15.8, 87.4]	29.3 [15.0, 94.2]		30.0 [17.8, 80.5]	33.0 [17.1, 82.3]	
Missing	2 (0.5%)	0 (0%)		NA	NA	
BMI						
Mean (SD)	18.5 (3.81)	18.4 (4.15)	0.819	18.3 (3.67)	18.7 (3.67)	0.442
Median [Min, Max]	17.8 [8.18, 34.0]	17.2 [12.8, 41.4]		17.0 [13.4, 34.3]	17.5 [14.2, 33.7]	
Missing	2 (0.5%)	0 (0%)		1 (1.4%)	0 (0%)	
High-Dose Oral Steroid Courses						
Mean (SD)	2.15 (0.388)	1.41 (0.776)	<0.001	2.14 (1.27)	0.0603 (0.239)	<0.001
Median [Min, Max]	2.00 [2.00, 5.00]	2.00 [0, 2.00]		2.00 [1.00, 6.00]	0 [0, 1.00]	
ED (emergency department) visits						
Mean (SD)	4.60 (4.19)	2.75 (3.00)	<0.001	0.451 (0.968)	0.0517 (0.222)	0.00102
Median [Min, Max]	4.00 [0, 30.0]	2.00 [0, 22.0]		0 [0, 5.00]	0 [0, 1.00]	
% Predicted Pre-BD FEV1						
Mean (SD)	91.9 (14.1)	109 (18.2)	<0.001	91.2 (13.0)	93.8 (13.0)	0.181
Median [Min, Max]	92.8 [31.8, 135]	111 [34.6, 154]		92.0 [62.0, 118]	95.0 [61.0, 125]	
Airway hyper-responsiveness						
Mean (SD)	0.671 (0.336)	0.998 (0.532)	<0.001	−0.260 (1.15)	0.162 (1.11)	0.0149
Median [Min, Max]	0.778 [0.043, 2.2]	0.778 [0.14, 2.87]		−0.390 [−2.59, 2.48]	0.140 [−2.94, 2.53]	
Bronchodilator Response as % of baseline FEV1						
Mean (SD)	6.57 (8.37)	3.30 (7.33)	<0.001	0.130 (0.248)	0.0764 (0.0731)	0.0805
Median [Min, Max]	5.29 [−18.1, 44]	3.40 [−35.1, 25]		0.0800 [−0.04, 2.05]	0.0600 [−0.06, 0.41]	
Log10 IgE						
Mean (SD)	2.61 (0.629)	2.40 (0.669)	0.0255	2.78 (0.622)	2.60 (0.669)	0.0684
Median [Min, Max]	2.73 [0.260, 3.7]	2.55 [0.73, 3.67]		2.89 [1.26, 4.13]	2.61 [0.30, 4.15]	
Missing	244 (64.4%)	118 (58.7%)		NA	NA	
25 Hydroxyvitamin D (ng/mL)						
Mean (SD)	36.0 (11.0)	35.0 (9.18)	0.484	39.9 (15.7)	40.2 (15.2)	0.902
Median [Min, Max]	34.7 [12.5, 71.5]	34.4 [18.4, 63.1]		37.8 [14.6, 80.0]	39.5 [9.00, 75.9]	
Missing	244 (64.4%)	118 (58.7%)		NA	NA	
Blood Eosinophils (Log10)						
Mean (SD)	2.63 (0.439)	2.52 (0.394)	0.00373	2.57 (0.462)	2.52 (0.427)	0.486
Median [Min, Max]	2.73 [−1.0, 3.30]	2.57 [1.0, 3.27]		2.67 [0, 3.40]	2.61 [0, 3.24]	
Missing	13 (3.4%)	6 (3.0%)		1 (1.4%)	2 (1.7%)	
Smoking						
No	281 (74.1%)	152 (75.6%)	0.59	46 (64.8%)	67 (57.8%)	0.508
Yes	97 (25.6%)	46 (22.9%)		25 (35.2%)	47 (40.5%)	
Missing	1 (0.3%)	3 (1.5%)		0 (0%)	2 (1.7%)	
Asthma Severity						
Mild	NA	NA		29 (40.8%)	51 (44.0%)	0.79
Moderate	NA	NA		42 (59.2%)	65 (56.0%)	
Genetic Ancestry Proportion	(*n* = 366)	(*n* = 194)		(*n* = 48)	(*n* = 83)	
Sub Saharan African	0.047	0.045	0.29	0.167	0.179	0.84
Central and South Asia	0.014	0.014	0.40	0.009	0.01	0.33
East Asia	0.003	0.003	0.70	0.002	0.003	0.18
Europe	0.533	0.539	0.26	0.772	0.734	0.52
Native America	0.320	0.317	0.61	0.030	0.045	0.35
Oceania	0	0	0.06	0	0	0.7
Middle East	0.082	0.080	0.3	0.019	0.027	0.15

#### 3.1.2. CAMP

Small RNA-Seq data on the baseline serum level were available for 492 (47%) of the 1041 children in the CAMP. A total of 187 CAMP participants (38%) who were on ICS and had enough data to calculate their CASTER phenotype were divided into two groups: those with CASTER values below the mean CASTER value (0.00547) were classified as poor responders (*n* = 116), and those with CASTER values above the mean CASTER value were classified as good responders (*n* = 71) (Table 1).

In GACRS, older age (9.43 vs. 8.8) was associated with poor ICS response (*p* < 0.001), whereas in CAMP, there was a non-significant trend of younger age associated with poor ICS response. Total serum IgE level and eosinophil count were higher in the poor-responder group than in the good-responder group in both cohorts (Table 1). Further, the five constituent features of the CASTER measure differed from poor to good ICS responders, as expected, described presently. The poor-responder group patients had more courses of oral steroids (GACRS: 2.15 vs. 1.41; CAMP: 2.14 vs. 0.06) and ED visits (GACRS: 4.6 vs. 2.75; CAMP: 0.451 vs. 0.052) as well as more reactive airways (GACRS: 0.671 vs. 0.99; CAMP: −0.26 vs. 0.16) compared to good responders. Poor responders similarly had higher BDR (GACRS: 6.57 vs. 3.30; CAMP: 0.13 vs. 0.076) than the good-responder group (Table 1). Poor responders had a lower FEV1 than good responders in the GACRS (91.9 vs. 109), but not in CAMP. We noticed no significant differences by asthma severity at enrollment (in CAMP; unavailable in GACRS) or by genetic ancestry proportions.

### 3.2. Identification of miRNAs Associated with ICS Response

The sequenced serum miRNA datasets were subjected to—quality control (QC), filtration, and normalization, yielding 580 samples and 317 miRNAs for our discovery cohort (GACRS) and 187 samples and 257 miRNAs in our replication cohort (CAMP). Logistic regression showed that 36 of the 317 interrogated miRNAs were associated with ICS response at 10% FDR in the GACRS (Figure 1 and Appendix A). Of these miRNAs, 33 were associated with poor ICS response (odds ratio (OR) >1) and three miRNAs were associated with good ICS response (OR < 1). Of these 36 miRNAs, three (miR-28-5p, miR-339-3p and miR-432-5p) were validated as being significant in the CAMP cohort and in the same direction of effect (Table 2A,B, Appendix A) at *p* < 0.05. All three replicated miRNAs were positively associated with poor ICS response (OR > 1, Table 2), which means that the increase in expression of these miRNAs leads to worse response to ICS treatment. In CAMP, the miRNA association finding remains consistent in sensitivity analysis adjusting by asthma severity status and genetic ancestry proportion.

In order to predict ICS response, three predictive models were assessed in this study. Each logistic regression model was trained on the GACRS cohort and tested on the CAMP cohort without parameter refitting. Model1, which incorporated demographic variables and clinical variables which were not constituents of the CASTER measure, had a 65% (CI: 57–73%) AUC in testing on the CAMP cohort (Appendix A). Model2, which included only the three replicated miRNAs, had a 64% (CI: 56–72%) AUC in the same cohort (Appendix A). Finally, Model3, which combined all variables from both Model1 and Model2, had a 72% (CI: 65–79%) AUC in the CAMP cohort (Appendix A). ANOVA was used to compare the models, and the results indicated a significant difference only between Model1 and Model3 (*p* = 0.0022).

### 3.3. Target Identification and Functional Enrichment Analysis of Replicated miRNAs

Next, we assessed the putative targets of these three miRNAs. We used the multiMiR R package to identify 1320 unique functionally validated gene targets for pathway and ontology enrichment (Table 3, Figure 2A). We found enrichment of some common asthma pathogenesis-associated pathways such as PI3K-Akt, MAPK cascade, Wnt, Hippo, FoxO, and p53 signaling. We also discovered enriched ICS-response-related pathways such as estrogen receptor (ESR)-mediated, PI3K-Akt signaling (Appendix A), and immune-response- and inflammation-associated pathways such as neutrophil degranulation and Interleukin-4 and -13 signaling (Figure 2B).

### 3.4. In Vitro Steroid Response Lymphoblastoid Gene Expression

Paired mRNA gene expression data were available from immortalized lymphoblastoid cell lines treated with dexamethasone (DEX) or a sham vehicle treatment from some of the CAMP subjects. From these, we selected 88 paired DEX–sham arrays from individuals for which serum miRNA data was also available. A total of 3827 out of 4818 probes passing QC were differentially expressed in response to DEX (complete list by module in Appendix A). These genes showed significant enrichment of direct targets of the three replicated miRNAs (miR-28-5p, miR-339-3p, and miR-432-5p; *p* = 0.042). Further, association analysis between three miRNAs and DEGs showed 22 significant associations at 10% FDR (Appendix A).

WGCNA analysis was then conducted on the 88 paired DEX–sham expression arrays to identify associations between gene expression on ICS response and three miRNAs associated with ICS response. We identified 20 gene modules showing coordinated response of LCL cells to dexamethasone treatment. Next, we calculated the module eigengene (ME) for each of the 20 gene modules and used linear mixed-effects models to evaluate the association between each of the gene modules with the three miRNAs. We found that the black (OR = 0.99, FDR = 0.02) and magenta (OR = 0.99, FDR = 0.01) modules (Figure 3, Appendix A) were significantly associated with miR-339-3p. The functional enrichment analysis of these two modules showed that the magenta module is enriched with genes associated with adaptive immune response and inflammation pathways such as adaptive immune response, PIP3 activates AKT signaling, PTEN regulation, mTOR signaling, and glutathione metabolism. We also looked for an association between these 20 gene module eigengene values and clinical and demographic outcomes, but we did not find any significant association at the 5% FDR cut-off.

## 4. Discussion

MicroRNAs are emerging as pharmacogenomic biomarkers capable of identifying drug treatment response [30]. By enhancing the reliability of asthma treatment regimens, a biomarker for ICS response might subsequently help reduce asthma morbidity. In this study, we used CASTER, a composite corticosteroid responsiveness phenotype, to identify serum miRNAs as indicators of ICS response in two independent cohorts of children with asthma. GACRS is a cross-sectional observational study with self-reported ICS use. CAMP, on the other hand, is a clinical trial in which all children in the ICS-arm received the same dose of the same ICS. Although GACRS is a cross-sectional study, and CASTER is designed to provide a cross-sectional measure of ICS response, this is still an inferred response phenotype that may differ from ICS response as revealed in a longitudinal cohort such as CAMP. We minimized such issues by also computing CASTER in the CAMP cohort, but some differences between the studies may have resulted in a lower number of replicated miRNAs.

We identified 36 miRNAs significantly associated with ICS response at 10% FDR cut-off; out of these, 17 miRNAs were found to have passed a more stringent 5% FDR cut-off, three of which (miR-28-5p, miR-339-3p and miR-432-5p) were found to be positively associated with poor ICS response in both cohorts (Appendix A). Even at 5% FDR, both miR-28-5p and miR-339-3p remained significant, while miR-432-5p was borderline, at *p* = 0.058 (Table 2B). These three miRNAs showed some promise as potential pharmacogenomic biomarkers for ICS response with a 64% AUC in a test cohort when evaluated alone (Appendix A), providing a significant improvement to prediction based solely on clinical and demographic data (72% AUC). While we stress that this performance is with no parameter refitting in a completely independent cohort, and is therefore very encouraging, further work would be required to obtain more comprehensive models suitable for clinical use.

miR-28-5p has been previously associated with inflammation in severe asthma [31]. miR-28-5p is a validated regulator of *IL2RG* (*interleukin 2 receptor, gamma*) (Appendix A), which is a pro-inflammatory cytokine receptor that plays an important role in glucocorticoid function in severe asthma [32,33]. miR-339-3p regulates the expression of *NR3C1*, which encodes an intracellular glucocorticoid receptor [34]. Fez et al. 2019 showed that miR-339-3p was up-regulated after 30 months of ICS treatment in COPD patients [35]. We are the first to report a link between miR-432-5p and asthma and ICS response, but this miRNA has previously been linked to cancer [36].

Functional enrichment analysis of their unique validated targets showed that these miRNAs regulate the expression of genes enriched in pathways of immune response and inflammation (Figure 2B). This may indicate a possible role of these miRNAs in immune dysregulation. The previous literature has established a link between immune dysregulation and steroid insensitivity in patients [32,37,38]. We also discovered that these miRNAs regulate the expression of genes that were enriched in estrogen receptor (ESR)-mediated (Appendix A) and PI3K-Akt signaling pathways (Appendix A), which have been linked to steroid resistance in severe asthma [39]. According to Bi et al. 2020, inhibiting the PI3K-Akt signaling pathway with PI3K inhibitors mitigates glucocorticoid (GC) insensitivity in severe asthma by restoring HDAC2 activity and inhibiting the phosphorylation of nuclear signaling transcription factors [40]. It has previously been reported that estrogen inhibits glucocorticoid anti-inflammatory function. [41,42]. This suggests that these miRNAs indirectly contribute to glucocorticoid insensitivity by regulating the genes involved in the pathways.

miR-339-3p is significantly associated with two DEX-responsive gene clusters (Figure 3). The genes in these modules were found to be enriched in inflammation-related pathways: adaptive immune response, PIP3 activates AKT signaling, PTEN regulation, mTOR signaling, and glutathione metabolism (Figure 4). miR-339-3p showed a significant association (FDR = 0.02) with a DEX-responsive gene (*PSMA7*) (FDR = 0.0023) which participates in activation of NF-kappaB in B cells (R-HSA-1169091) (Appendix A). NF-kappaB is a pro-inflammatory transcription factor which is well-studied for its role in glucocorticoid response [43,44]. miR-339-3p also showed significant association with genes of adaptive immune system (*CLTA*, *clathrin* at FDR = 0.02 and *RNF126*, *ring finger protein 126* at FDR = 0.04), mTOR-signaling pathway (*EIF4EBP1*, *eukaryotic translation initiation factor 4E binding protein 1* at FDR = 0.04; *CAB39*, *calcium binding protein 39* at FDR = 0.05), and HATs acetylate histones pathway (*RUVBL2* at FDR = 0.06). The mTOR signaling pathway produces inflammatory or immune imbalance in asthma [45], and it has been reported that inhibition of mTOR activity can restore corticosteroid sensitivity in COPD [46]. The HATs acetylate histones pathway is critical in corticosteroid responsiveness. Corticosteroids suppress inflammatory genes in asthma by inhibiting HAT activity and recruiting HDAC2 to the activated inflammatory gene complex. HDAC2 activity and expression are reduced in severe asthma, which may account for the increased inflammation and resistance to corticosteroid action [47,48].These findings suggest that these three miRNAs, particularly miR-339-3p, play an important role in immune dysregulation, which results in poor response to ICS treatment.

The advantages of our study include a relatively large population for integrative pharmacogenomics in asthma, replication of the results in another childhood asthma cohort, use of a previously validated composite ICS response phenotype, and validation of the function of selected miRNAs using in vitro steroid response lymphoblastoid cell line gene expression data. Disadvantages of our study are the lack of mRNA expression data for the patient at the same time point as the LCL expression data, as well as racial and trial-design differences between our discovery and replication cohorts. While CASTER may be thought of as a measure of asthma severity, it would be one defined only in terms of ICS response: people doing poorly on ICS therapy are generally defined as having more severe asthma according to modern GINA (Global Initiative for Asthma) guidelines [49]. However, asthma severity was assessed at enrollment in CAMP in a treatment-independent manner, and this was not associated with the CASTER measure.

## 5. Conclusions

In conclusion, our findings show that serum miRNA expression differs among individuals according to response to ICS treatment. The miRNAs miR-28-5p, miR-339-3p, and miR-432-5p were associated with poor ICS response and may form part of a future biomarker of ICS response. Particularly, miR-339-3p may reduce ICS responsiveness through immune dysregulation.

## Figures and Tables

**Figure 1 cells-12-01505-f001:**
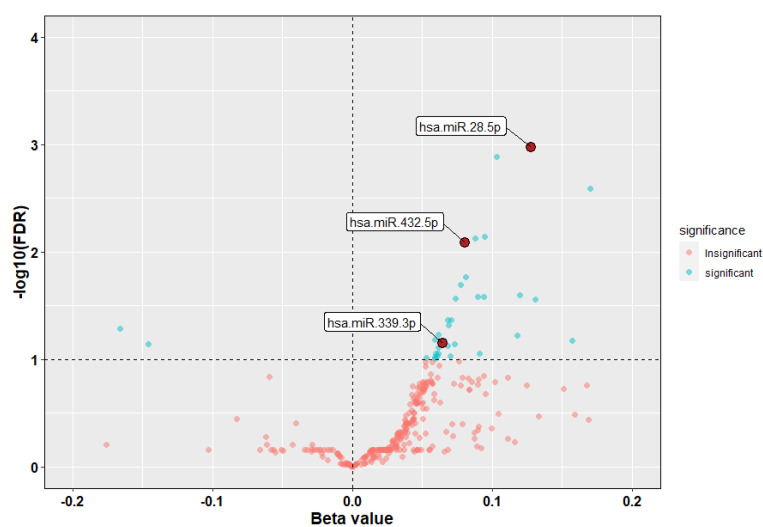
Serum miRNA association with CASTER in GACRS. Firebrick and cyan color circles indicate insignificantly and significantly associated miRNAs, respectively, and dark-red solid circles indicate miRNAs replicated in CAMP. Beta values are from regression analysis.

**Figure 2 cells-12-01505-f002:**
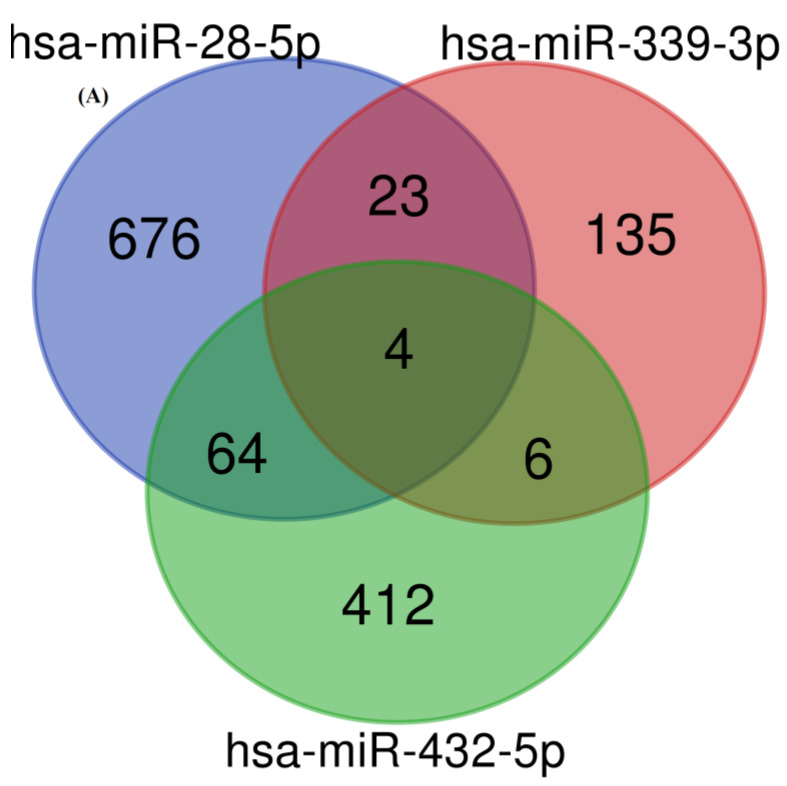
Putative differentially expressed miRNAs target genes. (**A**) Venn diagram showing the number of validated target genes of the 3 replicated DE-miRNAs. (**B**) Three miRNAs target genes enrichment analysis using DAVID version 2021 (gene ontology terms: molecular function and biological processes; pathway database: KEGG and Reactome) at 10% FDR cut-off.

**Figure 3 cells-12-01505-f003:**
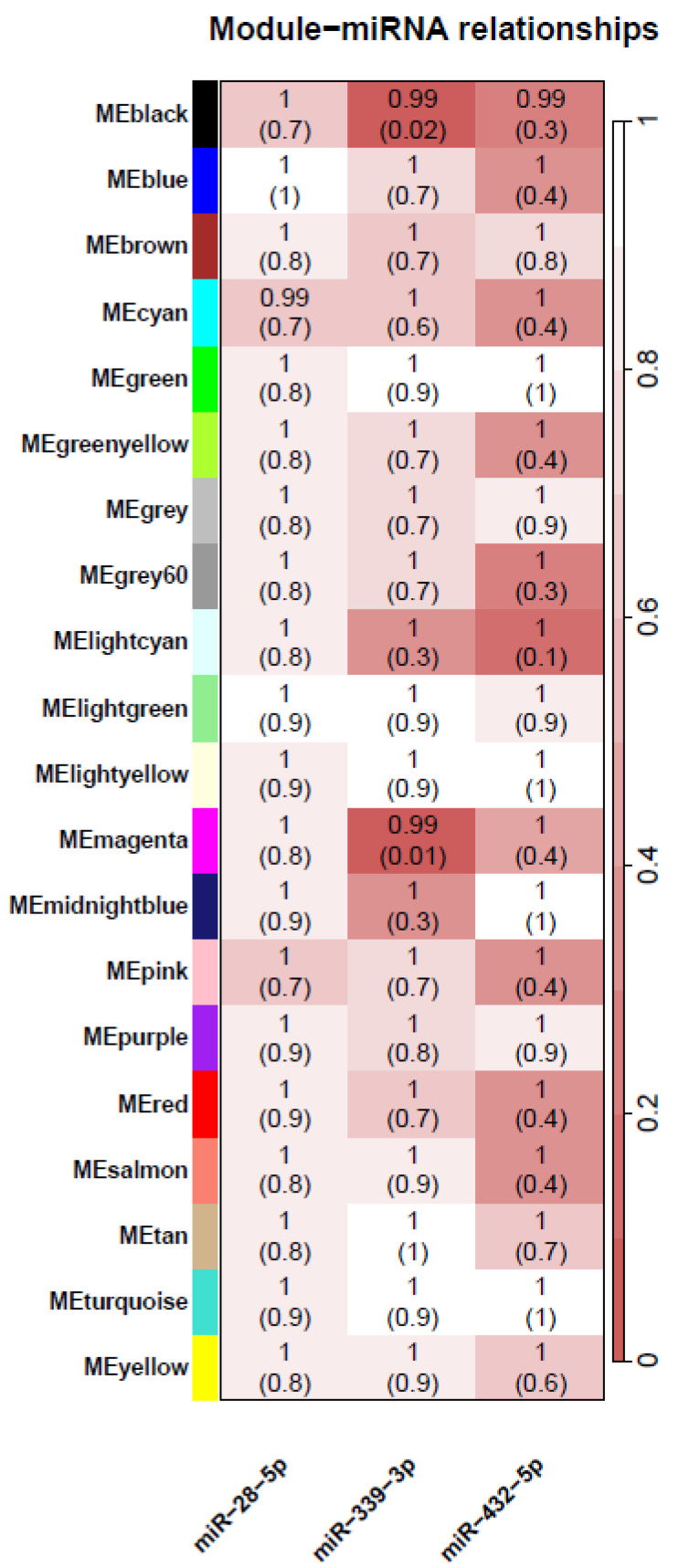
mRNA expression modules and replicated miR associations. Each row corresponds to a module eigengene of mRNA expression, columns to a replicated miR differentially expressed in poor vs. good ICS response. Each cell contains the corresponding odds ratio (OR) and adjusted *p*-value (FDR) from a linear mixed-effects model. The table is color-coded by adjusted *p*-value according to the color legend.

**Figure 4 cells-12-01505-f004:**
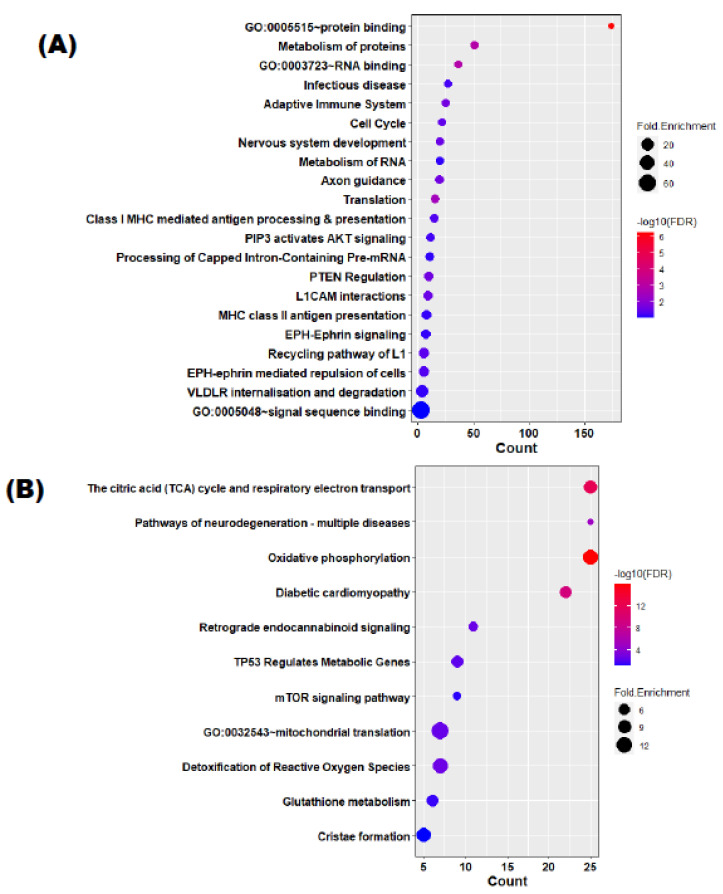
Black and magenta modules’ functional enrichment result. (**A**) The magenta module’s genes’ functional enrichment result; (**B**) The black module’s genes’ functional enrichment result. Performed using DAVID version 2021 (gene ontology terms: molecular function and biological processes; pathway database: KEGG and Reactome) at 10% FDR cut-off.

**Table 2 cells-12-01505-t002:** Significant miRNAs associated with CASTER (poor vs. good response) in GACRS and CAMP. Logistic regression model was adjusted for age and sex. *p* adjust: false discovery rate (FDR) adjusted *p*-values, with FDR < 0.10 considered significant.

**(A)**						
**Logistic Regression GACRS (Discovery Cohort)**						
**Variable**	**Beta**	**Z**	***p* Value**	***p* Adjust**	**OR**	**OR (95% CI)**
hsa-miR-339-3p	0.064	2.238	0.025	0.070	1.066	1.008–1.128
hsa-miR-28-5p	0.128	3.581	0.000	0.001	1.136	1.059–1.218
hsa-miR-432-5p	0.080	2.988	0.003	0.008	1.084	1.028–1.142
**(B)**						
**Logistic Regression CAMP (Replication Cohort)**						
**Variable**	**Beta**	**Z**	***p* Value**		**OR**	**OR (95% CI)**
hsa-miR-339-3p	0.199	2.620	0.009		1.221	1.052–1.417
hsa-miR-28-5p	0.172	2.408	0.016		1.188	1.033–1.366
hsa-miR-432-5p	0.128	1.898	0.058		1.137	0.996–1.298

**Table 3 cells-12-01505-t003:** Statistics of validated targets identified from microRNA-target databases (miRecords, miRTarBase and TarBase) for replicated DE-miR.

microRNAs Name	Number of Target Genes
hsa-miR-28-5p	767
hsa-miR-339-3p	168
hsa-miR-432-5p	486
Overall number of unique target genes	1320

## Data Availability

Sequencing data are available in Gene Expression Omnibus (accession pending).

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
