# Peer review of "Systems Genomics Reveals microRNA Regulation of ICS Response in Childhood Asthma"

_cells, 2023, doi:10.3390/cells12111505_

Round 1
Reviewer 1 Report
This is an interesting article on asthma patients' responses to inhaled corticosteroids (ICS). In this regard, the authors identified some circulating miRNAs that seem to play a role in the ICS response in childhood asthma. In particular, they analysed the expression of miRNAs in peripheral blood from 580 children with asthma treated with ICS and identified 36 microRNAs involved in ICS response.
The article is generally well written and structured. The introduction provides sufficient background and include all relevant references. However, I have a few observations:
1. Could you provide more details on the circulating microRNAs? In particular, can you explain the difference between extracellular/circulating microRNAs?
2. Please enter abbreviations the first time you use them (e.g WGCNA line 32)
3. Improve the presentation of Table 1 which is quite confusing
4. Improve the resolution of Figure 2
5. Provide a title for each figure and a detailed caption. Each figure has a very long title together with a caption. Separating things!
Also, the quality of the writing could have been much better.

Reviewer 2 Report
Major comments.
1. Asthma severity. A) Please detail asthma severity status and guidelines used to define asthma severity status in Table 1. B) CASTER may be capturing asthma severity while on ICS. Do the findings remain consistent in sensitivity analysis adjusting by asthma severity status?
2. Descriptives and covariate adjustment. The analysis were adjusted by age and sex (GARCS) and age, gender, and race/ethnicity (CAMP). A) Please confirm if biological sex or gender were used as a covariates? If biological sex was used, please refer it accordingly throughout the text. B) Please state clearly how “race”, and “ethnicity” was defined (e.g. See PMID: 36119389). Please note that Table 1 states “race” but not “ethnicity”. C) Once B is clarified, would it be relevant to evaluate differences in genetic ancestry proportions? ie. are there differences in genetic ancestry between poor and good responders? Do the findings remain consistent in sensitivity analysis adjusting by genetic ancestry proportions or principal components from the genotype matrix?
3. Why was a FDR threshold of 10% used and what is your power to detect significant signals at FDR of 10% and 5% in the discovery stage? Acknowledge how your results would have variated if a the most commonly used 5% threshold would have been used throughout all your analyses.
4. Predictive model. For miRNAs to be evaluated in terms of clinical utility, compare the results of three predictive models, including p-values ​​for the comparisons stated below. No figure regarding the predictive model is shown, despite it is mentioned in the text (A2). Please, include confidence interval for AUC.
a. Clinical prediction model (composite phenotype, with and without adjustment for relevant covariates (see 2).
b. Model that includes the 3 miRNAs
c. Clinical prediction model + miRNAs
5. WGCNA. Relevant additional insights could be gained from further consideration. A) Were the 20 module eigengene values associated with clinical and demographic outcomes (i.e. Table 1, plus genetic ancestry, if relevant). B) Describe the genes within each relevant module in a separate table. C) Were any of the relevant modules enriched in genes whose gene expression levels were regulated by the 3 miRNA. D) Were gene expression levels associated with clinical outcomes? If so, would causal inference analysis be relevant in this context?
Minor.
1. Open science. To increase transparency and adoption of open science practices, have the authors considered sharing the analytical pipeline and summary statistics on repositories (e.g., github, zenodo)
Round 2
Reviewer 2 Report
All the concerns have been addressed. Peer-review has resulted in a significant improvement of the manuscript. The authors acknowledge that the predictive capability of their findings still needs to be examined in other cohorts. However, it is promising to see some improvement in the clinical model after including the miRNA.